# Conventional PCR Versus Next Generation Sequencing for Diagnosis of FLT3, IDH and NPM1 Mutations in Acute Myeloid Leukemia: Results of the PETHEMA PCR-LMA Study

**DOI:** 10.3390/cancers17050854

**Published:** 2025-03-01

**Authors:** Blanca Boluda, Rebeca Rodriguez-Veiga, Claudia Sargas, Rosa Ayala, María J. Larráyoz, María Carmen Chillón, Elena Soria-Saldise, Cristina Bilbao, Esther Prados de la Torre, Irene Navarro, David Martinez-Cuadron, Cristina Gil, Teresa Bernal, Juan Bergua, Lorenzo Algarra, Mar Tormo, Pilar Martínez-Sanchez, Estrella Carrillo-Cruz, Josefina Serrano, Juan M. Alonso-Domínguez, Raimundo García, Maria Luz Amigo, Pilar Herrera-Puente, María J. Sayas, Esperanza Lavilla-Rubira, María José García-Pérez, Julia Morán, Esther Pérez-Santaolalla, Natalia Alonso-Vence, Ana Oliva, Juan Antonio López, Manuel Barrios, María García-Fortes, María Teresa Olave, Jorge Labrador, Joaquín Martínez-López, María J. Calasanz, Ramón García-Sanz, José A. Pérez-Simón, María T. Gómez-Casares, Joaquín Sánchez-Garcia, Yolanda Mendizabal, Eva Barragán, Pau Montesinos

**Affiliations:** 1Hematology Department, Hospital Universitari i Politécnic-IIS La Fe, 46026 Valencia, Spain; rebeca_rodriguez@iislafe.es (R.R.-V.); irene_navarro@iislafe.es (I.N.); david_martinez@iislafe.es (D.M.-C.); yolanda_mendizabal@iislafe.es (Y.M.); montesinos_pau@gva.es (P.M.); 2Molecular Biology Unit, Hospital Universitari i Politécnic-IIS La Fe, 46026 Valencia, Spain; claudia_sargas@iislafe.es (C.S.); barragan_eva@gva.es (E.B.); 3Hematology Department, Hospital Universitario 12 de Octubre, CNIO, Complutense University, 28041 Madrid, Spain; rayala@ucm.es (R.A.); mariapilar.martinez.sanchez@salud.madrid (P.M.-S.); jmarti01@med.ucm.es (J.M.-L.); 4CIMA LAB Diagnostics, Universidad de Navarra, 31008 Pamplona, Spain; mjlarra@unav.es (M.J.L.); mjcal@unav.es (M.J.C.); 5Hospital Universitario de Salamanca (HUS/IBSAL), CIBERONC and Center for Cancer Research-IBMCC (USAL-CSIC), 37007 Salamanca, Spain; mcchillon@saludcastillayleon.es (M.C.C.); rgarcias@usal.es (R.G.-S.); 6Hospital Universitario Virgen del Rocío, Instituto de Biomedicina (IBIS/CSIC/CIBERONC), Universidad de Sevilla, 41013 Sevilla, Spain; elena.soria@juntadeandalucia.es (E.S.-S.); estrellam.carrillo.sspa@juntadeandalucia.es (E.C.-C.); josea.perez.simon.sspa@juntadeandalucia.es (J.A.P.-S.); 7Hospital Universitario de Gran Canaria Dr. Negrín, 35010 Las Palmas de Gran Canaria, Spain; cbilsie@gobiernodecanarias.org (C.B.); mgomcasf@gobiernodecanarias.org (M.T.G.-C.); 8IMIBIC, Hematology, Hospital Universitario Reina Sofía, UCO, 14004 Córdoba, Spain; esther.prados@imibic.org (E.P.d.l.T.); josefina.serrano.sspa@juntadeandalucia.es (J.S.); joaquin.sanchez.garcia.sspa@juntadeandalucia.es (J.S.-G.); 9CIBERONC Instituto de Salud Carlos III, 28029 Madrid, Spain; 10Hospital General Universitario de Alicante, 03010 Alicante, Spain; gil_cricor@gva.es; 11Hospital Universitario Central de Asturias, Instituto Universitario (IUOPA), Instituto de Investigación del Principado de Asturias (ISPA), 33011 Oviedo, Spain; teresa.bernal@sespa.es; 12Hospital Universitario San Pedro de Alcántara, 10003 Cáceres, Spain; juanmiguel.bergua@salud-juntaex.es; 13Hospital Universitario General de Albacete, 02008 Albacete, Spain; jlalgarra@sescam.jccm.es; 14Hematology Department, Hospital Clínico Universitario-INCLIVA, 46010 Valencia, Spain; tormo_mar@gva.es; 15Hospital Universitario Fundación Jiménez Díaz, 28040 Madrid, Spain; juan.adominguez@fjd.es; 16Hospital Universitari General de Castelló, 12004 Castellón, Spain; garcia_rai@gva.es; 17Hospital Universitario Morales Messeguer, 300008 Murcia, Spain; marial.amigo@carm.es; 18Hospital Universitario Ramón y Cajal, 28034 Madrid, Spain; pherrera.hrc@salud.madrid.org; 19Hospital Universitari Dr. Peset, 46017 Valencia, Spain; sayas_mjo@gva.es; 20Complexo Hospitalario Lucus Augusti, 27003 Lugo, Spain; esperanza.lavilla.rubira@sergas.es; 21Complejo Hospitalario Torrecardenas, 04009 Almería, Spain; mariaj.garcia.perez.sspa@juntadeandalucia.es; 22Hospital U. Puerta del Mar, 11009 Cadiz, Spain; juliamorsan@gmail.com; 23Hospital Donostia, 20014 Donostia, Spain; esther.perezsantaolalla@osakidetza.eus; 24Hospital Santiago de Compostela, 15706 Santiago, Spain; natalia.alonso.vence@sergas.es; 25Hospital Universitario Nuestra Señora de Candelaria, 38010 Tenerife, Spain; aoliher@gobiernodecanarias.org; 26Hospital General Ciudad de Jaen, 23007 Jaen, Spain; jantonio.lopez.lopez.sspa@juntadeandalucia.es; 27Hospital Carlos Haya, 29010 Málaga, Spain; manuel.barrios.garcia.sspa@juntadeandalucia.es; 28Hospital U. Virgen de la Victoria, 29010 Málaga, Spain; maria.garcia.fortes.sspa@juntadeandalucia.es; 29Hospital Clínico U. Lozano Blesa, 50009 Zaragoza, Spain; tolave@salud.aragon.es; 30Department of Hematology, Hospital University Burgos, 09006 Burgos, Spain; jlabradorg@saludcastillayleon.es

**Keywords:** acute myeloid leukemia, PCR, NGS (next generation sequencing)

## Abstract

This PETHEMA PCR-LMA study aimed to evaluate whether mutations detected by NGS (VAF cut-off of ≥5%) correlate with NPM1, FLT3-ITD, FLT3-TKD, IDH1, and IDH2 mutations detected using conventional PCR (analytical sensitivity 3%) in a nationwide network of seven reference laboratories. Between 2019 and 2021, 1685 adult acute myeloid leukemia (AML) patients with at least one centralized sample (NGS or PCR) at primary diagnosis or relapse/refractory episode were included, and 1288 paired NGS/PCR samples were analyzed. Considering PCR the gold-standard, for NPM1 NGS sensitivity was 98.5% and specificity 98.9%, for FLT3-ITD 73.8% and 99.6%, for FLT3-TKD 84.5% and 99.3%, for IDH1 98.7% and 98.7%, and for IDH2 99.1% and 97.7%, respectively. Overall, median days from sample reception until report were 7 for PCR and 28 for NGS. This study shows high concordance between NPM1 and IDH results using PCR and NGS. However, sensible important discrepancies are observed for FLT3 mutations.

## 1. Introduction

In recent years, the application of the NGS (next generation sequencing) to the study of patients with acute myeloid leukemia (AML) has revealed new molecular alterations that have been incorporated into diagnosis classification and risk stratification systems [1,2,3,4]. The current AML diagnostic algorithm requires a rapid knowledge of molecular alterations to start tailored therapy, including targeted therapies, and an exhaustive molecular profile to stratify patients and refine post-remission treatment [5,6]. Nowadays, the genetic profile has been integrated into day-to-day clinical decision making, and molecular laboratories are under pressure to obtain results in a timely manner to develop a personalized management approach [7]. However, conventional PCR methods based on gene-to-gene analysis have proven to be inefficient, and clinical laboratories have had to adapt technologically, complementing PCR-based assays with NGS to address the growing number of clinically relevant variants [8,9]. Despite the critical role that molecular genetic testing plays in AML treatment, several factors (methodological, geographical, age of patient, and cost) determine differences in access to molecular diagnosis for AML patients [10]. Given the complexity of the current molecular diagnostic approach to AML diagnosis, the Spanish Programa Español para Tratamientos en Hematología) PETHEMA group implemented a nationwide network of seven reference laboratories in order to provide a rapid and standardized molecular diagnosis for adult patients with AML [11].

The PCR-LMA study aimed to consolidate the PETHEMA nationwide centralized diagnostic and monitoring platform (PLATAFO-LMA), focusing on providing timely molecular diagnoses for adult patients with AML, prioritizing potentially druggable mutations (i.e., *NPM1*, *FLT3*, and *IDH*). Another goal was to evaluate whether mutations detected by NGS correlate with mutations detected using conventional PCR.

## 2. Materials and Methods

### 2.1. Platform and Reference Laboratorios

The PETHEMA (Programa Español de Tratamientos en Hematología) co-operative group established a nationwide NGS diagnostic platform (PLATAFO-LMA) composed of seven reference laboratories: the Hospital Universitario La Fe (HULF, Valencia, Spain), the Hospital Universitario de Salamanca (HUS, Salamanca, Spain), the Hospital Universitario 12 de Octubre (H12O, Madrid, Spain), the Hospital Universitario Virgen del Rocío (HUVR, Sevilla, Spain), the Hospital Universitario Reina Sofía (HURS, Córdoba, Spain), the Hospital Universitario de Gran Canaria Dr. Negrín (HUDN, Las Palmas de Gran Canaria, Spain), and CIMA LAB Diagnostics (UNAV, Pamplona, Spain).

The platform strategy consisted of rapid conventional PCR screening for actionable genes (*NPM1*, *FLT3*, *IDH1* and *IDH2*) followed by an NGS consensus myeloid panel to complete molecular characterization (*ASXL1*, *CALR*, *CBL*, *CEBPA*, *CSF3R*, *DNMT3A*, *EZH2*, *FLT3*, *GATA2*, *IDH1*, *IDH2*, *JAK2*, *KIT*, *KRAS*, *MPL*, *NPM1*, *NRAS*, *PHF6*, *PTPN11*, *RUNX1*, *SETBP1*, *SF3B1*, *SRSF2*, *TET2*, *TP53*, *U2AF1*, *WT1*, and *ZRSF2*).

### 2.2. Patients

From October 2019 to April 2021, unselected consecutive patients with AML at diagnosis, resistance, or relapse were registered from 71 PETHEMA institutions and were included in the PCR-LMA non-interventional study (NCT04446741). Bone marrow samples were collected at diagnosis or at relapse/refractory AML, and shipped to central PLATAFO-LMA laboratories. The Institutional Ethics Committee for Clinical Research approved this study. Written informed consent in accordance with the recommendations of the Declaration of Human Rights, the Conference of Helsinki, and institutional regulations were obtained from all patients.

### 2.3. Molecular Studies

#### 2.3.1. Single Gene Testing Methods

Screening for *NPM1* and *FLT3*-ITD mutational status were assessed using capillary electrophoresis following methods previously described by Gale et al. [12] and Thiede et al. [13]. The analytical sensitivity was 3%. Samples with positive results from fragment analysis were analyzed by Sanger sequencing to determine the *NPM1* type.

For *FLT3*-TKD2 (Asp835/Ile836), *IDH1* (Arg132), and *IDH2* (Arg140/Arg172) mutation detection, a High Resolution Melting (HRM) method was designed using primers described by Yamamoto et al., 2001 [14] and Tefferi et al., 2010 [15]. The analytical sensitivity was 3%.

#### 2.3.2. NGS Methodology

The adopted NGS strategy was previously described in [11]. *ASXL1*, *CEBPA*, *FLT3*, *IDH1*, *IDH2*, *NPM1*, *RUNX1*, and *TP53* were considered clinically relevant and mandatory for their implication in clinical guidelines, clinical trials, and risk stratification, and they were necessarily assessed in all cases. Other genes related to AML pathogenesis were also recommended, although their study was not performed for all network participants (*BRAF*, *CALR*, *CBL*, *CSF3R*, *DNMT3A*, *ETV6*, *EZH2*, *GATA2*, *JAK2*, *KIT*, *KRAS*, *MPL*, *NRAS*, *PTPN11*, *SETBP1*, *SF3B1*, *SRSF2*, *TET2*, *TP53*, *U2AF1*, *WT1*, and *ZRSF2*). Analytical sensitivity was variant allele frequency (VAF 1%), although the co-operative group established VAF ≥ 5% for variant reporting as clinically relevant.

External Quality Control:

The diagnostic platform established a quality control assay by exchanging control samples among reference laboratories every 9–12 months. To date, four cross validation rounds have been performed to ensure homogeneous results between laboratories [16].

Furthermore, in cases where officially recognized quality-control programs exist (UK NEQAS and others) for the different molecular determinations included in the platform, laboratories are recommended to participate in them.

### 2.4. Study End-Points and Statistical Analyses

The study end-points were: (1) frequency of mutations detected by conventional PCR, including *NPM1*, *FLT3*-ITD, *FLT3*-TKD, *IDH1*, and *IDH2*, (2) frequency of mutations detected by NGS panel (mandatory genes as described), (3) sensitivity and specificity of NGS for *NPM1*, *FLT3*-ITD, *FLT3*-TKD, *IDH1*, and *IDH2*, with conventional PCR used as a reference test, and (4) turn-around time for the diagnostic report (conventional PCR and NGS, at initial diagnosis and at R/R status.

Sensitivity is defined as the ability of a test to detect a positive result. The specificity of a test is defined as its ability to designate a sample which does not have a mutation as negative. Considering PCR as the orthogonal method, the sensitivity of NGS was assessed as the ratio between positive results in samples with NGS (using VAF ≥ 5% and ≥1%) and positive results with PCR; specificity was calculated as the ratio between negative results in NGS and negative results with PCR. The concordance rate of positive results was calculated by dividing the number of agreement results by the total number of positive results.

Turnaround time was calculated as the time from the sample’s receipt by the central laboratory to complete reports uploaded to the web platform (which will automatically email the report to the referral site).

Frequencies, means and deviations, and medians and intervals were used to describe the frequency of each mutation determined by PCR and NGS.

## 3. Results

From October 2019 to April 2021, 2009 Spanish AML patients were registered in the PETHEMA AML registry (NCT02607059), with 1685 (84%) patients from 71 institutions with at least one sample (NGS or PCR) available. Overall, 1411 PCR samples and 1671 NGS samples were included in the analyses, with 1288 paired NGS/PCR samples (1094 corresponding to diagnosis of AML, 103 to relapse and 88 to refractory AML).

### 3.1. Patient Baseline Characteristics

Among 1512 patients with an available sample at diagnosis, the median age was 65 years (range 19–98), and 57% (n = 860) were male. The median leucocyte count at diagnosis was 8.6 × 109/L (range 0.1–401), and 79% had ECOG < 2 (Table 1).

### 3.2. Frequency of NPM1, FLT3-ITD, FLT3-TKD, IDH1, and IDH2 Mutations by PCR

Among 1411 PCR samples analysed, 1200 corresponded to AML diagnosis, and 211 to R/R episodes.

NPM1 mutational status was determined in 1257 (89%) samples, and was not tested in 46/1200 (3.8%) samples at diagnosis vs. 46/211 (21.8%) in R/R status. Overall, 22% (N = 276) of samples were positive, 22.4% (258/1154) at diagnosis vs. 17.5% (18/103) at R/R episode (*p* = 0.25). The NPM1 mutation types were A (c.860_836dup863dup) (56%), B (c.863_864insCATG) (4.3%), D (c.863_864insCCTG) (3.1%), atypical (6.2%), and not available (29.5%).

FLT3-ITD mutation was tested in 1386 (98%) samples. The frequency of FLT3-ITD was 17.2% (N = 239), 17.2% (203/1181) at diagnosis, and 18.9% (36/192) at R/R episode.

FLT3-TKD mutation was tested in 1220 samples (86%). The frequency of FLT3-TKD was 5.1% (N = 62), 5% (55/1026) at diagnosis, and 3.7% (7/187) at R/R episode.

IDH1 mutation was tested in 1031 (73.1%) samples. The frequency of IDH1 mutation was 8.2% (N = 85), 7.2% (61/841) at diagnosis, and 12.7% (22/173) at R/R episode.

IDH2 mutation was tested in 1032 (73.1%) samples. The frequency of IDH2 mutation was 11.1% (N = 115), 11.6% (99/851) at diagnosis, and 8.4% (14/172) at R/R episode.

### 3.3. Frequency of Mutations Detected by NGS

The NGS mutational profile was available in 1671 samples, showing that almost all samples (96.5%) had at least one of the studied genes mutated, namely the most frequently mutated genes (frequency > 15%) FLT3 (both) (24.6%), NPM1 (22.7%), DNMT3A (22.5%), TET2 (20.6%), RUNX1 (18.3%), NRAS (17.9%), TP53 (17.9%), SRSF2 (16.6%), FLT3-ITD (16%), and IDH2 (15.3%) (Figure 1).

### 3.4. Comparison Between PCR and NGS

Among the 1288 paired NGS/PCR samples, NPM1 was available in 1285 (99.7%) for NGS and 1171 (90.8%) for PCR. Both results were available in 1156 (89.8%). Using a VAF cut-off of 5% for NGS, 23.5% were positive (272/1156) with a median VAF of 35.5% (range 7.1–74.9). The comparison of the positive results for each mutation for both techniques with VAF cut-off of ≥5% are shown in Figure 2. The NGS sensitivity was 98.5% (262/266) and specificity wad 98.9% (880/890) (Table 2 and Figure 3). Using a cut-off of ≥1% for NGS, 24% were positive for NPM1 (278/1156), with a sensitivity of 99.2% (264/266) and a specificity of 98.4% (876/890) (Table 3).

FLT3-ITD was available in 1284 (99.7%) for NGS and 1272 (98.8%) for PCR, with both available in 1267 (98.4%). With NGS VAF cut-off ≥ 5%, 12.1% were positive (153/1267) with a median VAF of 29.6 (range 5.21–97.4). By conventional PCR (cut-off ≥0.03), 15.9% were positive (202/1267), with a median ratio of 0.61 (range 0.03–11.6). We observed an NGS sensitivity of 73.8% and specificity of 99.6% (Table 2). Using a VAF cut-off ≥1%, the sensitivity increased to 87.1% (176/202) and specificity was 99% (1054/1065) (Table 3).

FLT3-TKD was available in 1283 (99.6%) for NGS and 1137 (88.3%) for PCR, with both available in 1130 (87.7%). With NGS VAF cut-off ≥5%, 5% were positive (57/1130) with median VAF 24.7 (range 5.06–51.8), resulting in sensitivity 84.5% and specificity 99.3% (Table 2). With VAF cut-off ≥1%, 7.7% were positive for FLT3-TKD (87/1130), with sensitivity of 91.4% (53/58) and specificity of 96.8% (1038/1072) (Table 3). Table 4 shows the FLT3-TKD alterations identified by NGS (cut-off ≥1%) and not detected by PCR.

IDH1 was available in 1283 (99.6%) for NGS and 929 (72.1%) for PCR, with both available in 925 (71.8%). With an NGS VAF cut-off of ≥5%, 9.5% were positive (88/925) with a median VAF of 41.8 (range 5.7–51.6), resulting in sensitivity and specificity of 98.7% (Table 2). With a VAF cut-off of ≥1%, 10.3% were positive for IDH1 (95/925), with a sensitivity of 98.7% (73/78) and a specificity of 97.9% (829/847) (Table 3).

IDH2 was available in 1285 (99.8%) for NGS and 931 (72.3%) for PCR, with both available in 931 (72.3%). With an NGS VAF cut-off of ≥5%, 13.5% were positive (126/931) with a median VAF of 42.2 (range 5.6–92.6), resulting in a sensitivity of 99.1% and a specificity of 97.7% (Table 2). With a VAF cut-off of ≥1%, 13.9% were positive for IDH2 (129/931), with a sensitivity of 99.7% (107/108) and a specificity of 97.3% (804/823) (Table 3).

The concordance rate between both techniques using NGS cut-off ≥5% was as follows: 95% for NPM1, 72% for FLT3-ITD, 74% for FLT3-TKD, 87% for IDH1, and 84% for IDH2 (Figure 4). The concordance rate between both techniques using an NGS cut-off of ≥1% were as follows: 95% for NPM1, 83% for FLT3-ITD, 58% for FLT3-TKD, 80% for IDH1, and 82% for IDH2 (Figure 5).

### 3.5. Comparative Between PCR and NGS at AML Diagnosis

Overall, 1094 patients were analyzed at diagnosis, NPM1 was available in 1091 (99.7%) for NGS and 1074 (98.2%) for PCR, with both results available in 1061 (97%). Using a VAF cut-off of ≥5%, 24% were positive with a median VAF of 35.8% (range 7.1–74.9).

FLT3-ITD was available in 1091 (99.7%) for NGS and 1084 (99%) for PCR, with both available in 1079 (98.5%). With an NGS VAF cut-off of ≥5%, 11.8% were positive with median VAF of 29.6% (range 5.27–97.4), and 15.8% were positive by PCR using a cut-off 0.03 ratio (median ratio 0.59; range 0.03–11.6).

FLT3-TKD was available in 1090 (99.5%) for NGS and 958 (87.6%) for PCR, with both results available in 951 (86.9%). With an NGS VAF cut-off of ≥5%, 5.2% were positive with a median VAF of 24.7% (range 5.1–51.8).

IDH1 was available in 1090 (99.5%) for NGS and 769 (70.3%) for PCR, with both available in 766 (70%). With an NGS VAF cut-off of ≥5%, 8.6% were positive with a median VAF of 41.3% (range 5.7–51.6).

IDH2 was available in 1092 (99.7%) for NGS and 771 (70.4%) for PCR, with both available in 771 (70.5%). With an NGS VAF cut-off of ≥5%, 14.3% were positive with a median VAF of 42.5% (range 6.2–92.6), NGS sensitivity and specificity for different genes are shown in Appendix A (using VAF cut-off of ≥5%) (Appendix A).

### 3.6. Comparative Between PCR and NGS at R/R AML Episode

Overall, 192 samples were tested at different R/R AML episodes. NPM1 was available in 191 (100%) for NGS and 96 (50%) for PCR, and for both in 95 (49.5%).

FLT3-ITD was available in all samples for NGS and 188 (97.9%) for PCR, and for both in 188 (97.9%). In positive cases, the median NGS VAF was 29.9% (range 5.2–92.6) and the median PCR ratio was 0.78 (range 0.03–11).

FLT3-TKD was available in all samples for NGS and 181 (93.8%) for PCR, and for both in 179 (93.2%).

IDH1 was available in all samples for NGS and 160 (82.9%) for PCR.

IDH2 was available in all samples for NGS and 160 (82.9%) for PCR. NGS sensitivity and specificity for different genes are shown in Appendix A (using cut-off VAF ≥ 5%) (Appendix A).

### 3.7. Turnaraound Time

Overall, the median days until full PCR report were 7 (range 0–84/interquartile range 3–14), and 28 (range 5–80/interquartile range 21–38) for NGS. At AML primary diagnosis, the median days until full PCR report were 5 (range 0–84/interquartile range 2–8), and 28 (range 5–80/interquartile range 20–38) for NGS. At R/R AML episodes, the median days until full PCR report were 12 (range 0–81/interquartile range 6–20), and 30 (range 11–74/interquartile range 24–41) for NGS (Appendix A).

## 4. Discussion

This study is the first national large study of a wide range of patients, comparing the results of NGS and PCR performed via a uniform process for the detection of relevant genes in an AML clinical setting. We show high concordance between *NPM1*, *IDH1*, and *IDH2* mutations detection using conventional PCR or NGS for AML patients. However, there were remarkable discrepancies between PCR and NGS to detect *FLT3*-ITD and *FLT3*-TKD mutations. The NGS approach showed several weaknesses to detect *FLT3*-ITD mutations with large duplications, whereas non canonical *FLT3*-TKD mutations and variants with low alleles were missed by PCR. On the other hand, the turnaround time for *NPM1*, *IDH*, and *FLT3* mutations was more rapid using conventional PCR (7 days) as compared to NGS (28 days). Our national diagnostic platform allowed for rapid access to potential targeted therapies using conventional PCR, and for full classification of AML and risk stratification for post-remission therapy using NGS.

The consolidation of the national diagnostic platform has definitively facilitated the access of AML patients to precision medicine. This new strategy has meant an increase in the quality of molecular studies at the clinical level and also in the field of translational research. In fact, in our study period (2019–2021) we found that 84% of registered patients had a centralized NGS, while the centralized PCR rate was 80%. These data contrast with our recently published REALMOL study analyzing treatment patterns of the AML diagnostic [17], where up to 95% of Spanish patients had NGS and PCR at diagnosis between 2017 and 2021. These differences reflect that a percentage of patients had only local diagnostic tests, especially for conventional PCR. Another clear area of improvement is the R/R AML episode, as presumably only a small fraction of episodes were centrally tested by NGS and/or PCR.

Our study cohort of 1512 patients was representative of AML epidemiology, but with a bias towards younger patients included (median age 65 years old), in line with the REALMOL study that showed that patients aged 70 years or more had less frequent diagnostic tests [17]. Nevertheless, the frequency of detected mutations by conventional PCR at diagnosis reflect a real world AML cohort (22% *NPM1*, 17% *FLT3*-ITD, 5% *FLT3*-TKD, *IDH1* 7%, and *IDH2* 11%) [18]. Of note, NGS and PCR testing at diagnosis for *NPM1* and *FLT3*-ITD were performed in parallel for 99% of patients, but *IDH1/2* genes were predominantly tested by NGS (99%) and less frequently by PCR (70%). These patterns may reflect a lack of tailored induction therapies for newly diagnosed *IDH1/2* AML during the study period, in contrast with *NPM1* and *FLT3* patients where rapid PCR screening was required to potentially add gentuzumab ozogamicine [19,20,21] or midostaurin [22], which could be to 3 + 7 chemotherapy.

We found that *NPM1* and *IDH2* mutations were less prevalent in R/R patients, indicating that *NPM1* AML [1], and even some *IDH2* types (i.e., R172) [5] could have a lower relapse rate, and that *IDH2* affects older patients [16,23] that are less frequently tested at R/R episodes. Conversely, we found that *FLT3*-ITD and *IDH1* mutations were more frequent at R/R, supporting the idea that these leukemias could be more resistant [1] (although the prognostic impact of *IDH1* remains controversial) [24,25,26]. Interestingly, we showed that NGS was performed more frequently than conventional PCR at R/R episodes for *NPM1* and *IDH* mutations. The lower rate of PCR performed at R/R suggests that the rapid diagnosis of these mutations was much less required in clinical practice, as no targeted therapies were available in our country beyond clinical trials. In fact, PCR was as frequent as NGS for the *FLT3* R/R episode diagnosis, influenced by gilteritinib accessibility [27,28,29].

The comparison between NGS and PCR performance showed a high concordance rate between *NPM1* and *IDH* results. This suggests that both techniques would be suitable for a precise molecular diagnosis [30]. NGS was able to detect between 4 and 15% (5–19% using VAF ≥1% cut-off) of additional *NPM1* or *IDH* mutations, respectively, which could be interpreted as a technique with a better sensitivity for the assessment of these alterations [7,9]. Nevertheless, the current timelines for NGS reporting indicate that, in our context, a more rapid screening for these mutations can be systematically provided by conventional PCR. On the other hand, for *FLT3*-TKD, NGS missed nine positive samples using a cut-off of 5%, probably because the sensitivity of PCR is 3%. Lowering the cut-off to 1% allowed additional positive samples to be identified. For the purposes of this study we consider these as true positives, and from a clinical point of view the FLT3 inhibitor would be included as part of the treatment. This is in line with the advantage associated with NGS in detecting a wider range of mutations compared to PCR detecting only canonical mutations in I836 and D835. Another NGS advantage is the accuracy to identify subclonal variants with low allele frequency. Our study showed up to 12% and 37% of *FLT3*-TKD mutations found only by NGS using VAF ≥ 5% and ≥1% cut-offs, respectively. Therefore, we consider that NGS should replace PCR for *FLT3*-TKD and also *IDH* testing if the results delivery timelines are shortened for the purposes of target therapy establishment.

Regarding *FLT3*-ITD detection, we found a low sensitivity of NGS due to the limitation that this technique has for detecting duplications with long fragments [31,32,33]. The use of a VAF ≥ 1% cut-off can improve detection efficiency but 12% of false negative results still persisted with NGS. As a consequence, the detection of *FLT3*-ITD mutation by NGS could be underestimated [34]. Therefore, we believe that nowadays NGS still cannot replace PCR for *FLT3*-ITD screening, since missing these mutations could impact patient stratification for FLT3 inhibitors (e.g., midostaurin, gilteritinib).

In this study, we have employed two different VAF cut-offs for NGS detection in order to explore performance outcomes, which result in noticeably different results. For these mutations with a high clinical impact, we consider these results as true positives and they are thus informed and considered. We recommend adopting the 1% cut-off as the standard for these mutations and their clinical implications.

The median time from sample receipt until full report delivery was 7 days for PCR and 28 days for NGS. This timing could be more or less influenced by shipment distance from referral site to central laboratory, which is estimated in 24 h in Spain. However, courier logistical issues or sample drawing on Friday (with delivery on Monday) could lead to substantial additional delays affecting centralization applicability. It should be noted that we captured the date of the full diagnostic report, while central labs were routinely offering partial results (e.g., *FLT3* positive or negative only) once this was available. Overall, our PCR and NGS timelines were in line with those recommended by the European Leukemia Net panel of experts (3–5 days for *NPM1*, *FLT3*, and *IDH* mutations, and before the end of the first cycle for remaining mutations by NGS) [1]. Interestingly, we show delayed timelines for molecular reporting in R/R AML, which may obey the general characteristics of R/R patients (usually less hyperleukocytosis and urgency to start therapy).

Our study has been carried out in a context in which molecular diagnosis laboratories in AML still need to combine conventional PCR with NGS. We can imagine a not-too-distant future in which, for the sake of efficiency, NGS will replace conventional PCR, at least for the vast majority of relevant mutations. The use of NGS for routine clinical care will require new NGS technologies (e.g., nanopore sequencing), more robust bioinformatic tools to improve detection of FLT3-ITD mutations, the ability to detect a wide range of alterations (including rearrangements i.e., MLLr) [35], and automated workflow to deliver a rapid turnaround time. These technological advances could pave the way to have a fully NGS-based molecular diagnosis in acceptable timelines (<5–7 days), allowing for the urgent personalized medicine treatment of AML. We could hypothesize that in a local laboratory with fewer samples it would be difficult to obtain these NGS timelines due to associated costs, while the centralization of samples would make this strategy affordable.

## 5. Conclusions

In conclusion, we show that a centralized platform at the national level is capable of providing molecular results by PCR and NGS according to the currently recommended timelines. While PCR remains the standard for the rapid screening of mutations in NPM1, IDH, and FLT3, NGS is capable of detecting more variants, especially in FLT3-TKD and IDH genes, which could have an impact on therapeutics. For these mutations with high clinical impact, a VAF cut-off of 1% should be adopted. Efforts should be made to report NGS results in acceptable times (<5–7 days) that may allow conventional PCR to be replaced and the daily routine in diagnostic laboratories to be optimized.

## Figures and Tables

**Figure 1 cancers-17-00854-f001:**
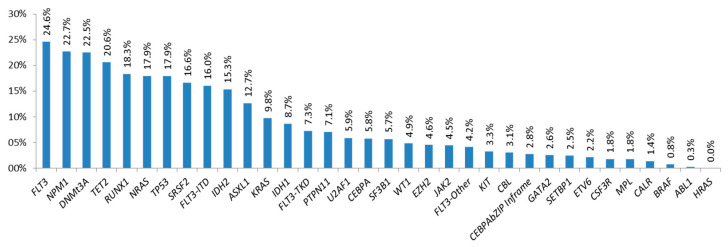
Frequency of mutations at diagnosis by NGS.

**Figure 2 cancers-17-00854-f002:**
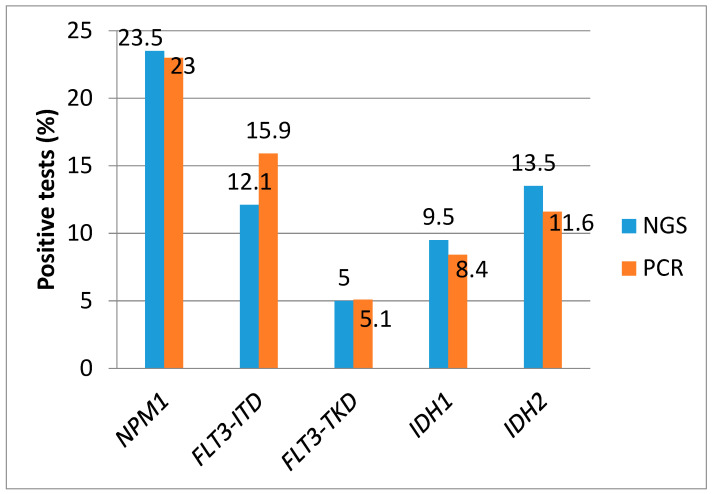
Positive results by conventional PCR and NGS (VAF cut-off ≥5%).

**Figure 3 cancers-17-00854-f003:**
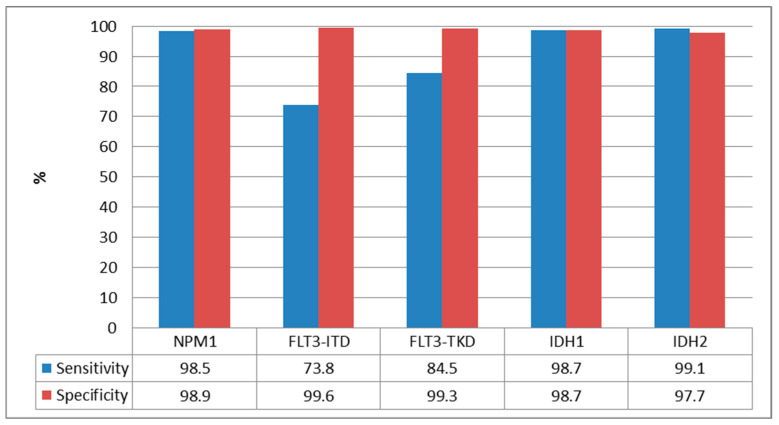
Sensitivity and specificity of NGS (VAF cut-off ≥5%) with respect to PCR.

**Figure 4 cancers-17-00854-f004:**
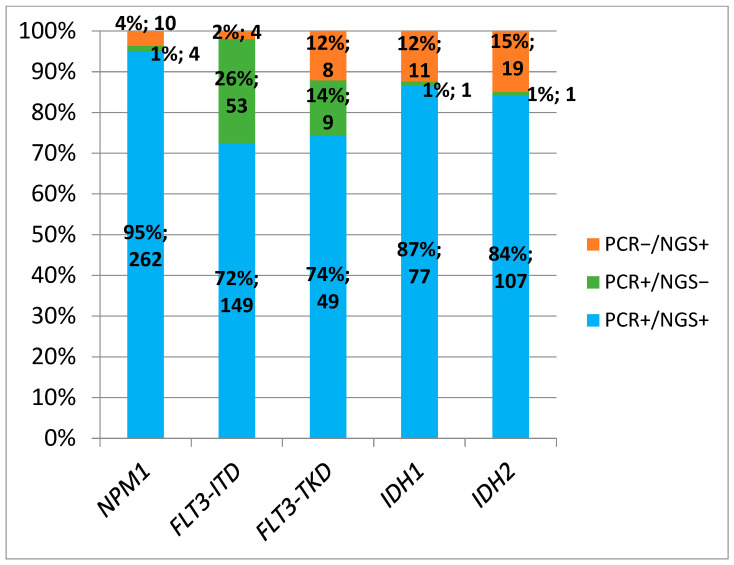
Concordance of positive results between PCR and NGS (VAF cut-off ≥5%).

**Figure 5 cancers-17-00854-f005:**
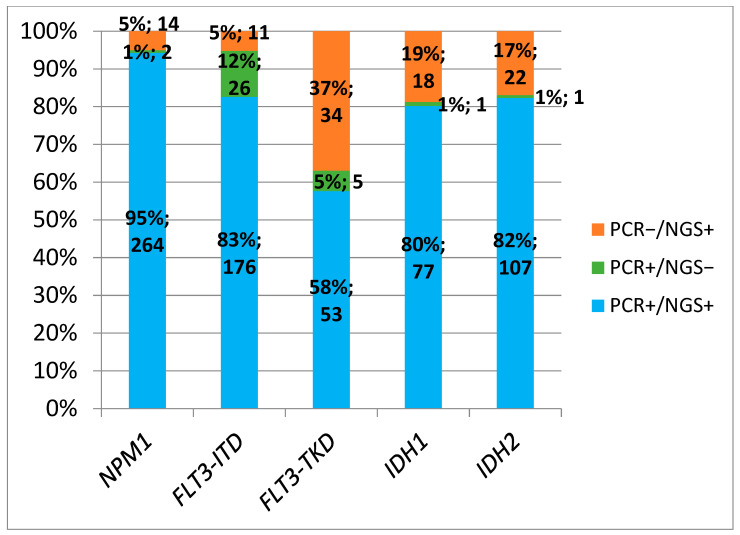
Concordance of positive results with PCR and NGS (VAF cut-off ≥1%).

**Table 1 cancers-17-00854-t001:** Main demographic and clinical characteristics of the study cohort.

Characteristic	Median (Range)	Number (%) n = 1512
Age at diagnosis	65 (19–98)	
<60 years		511 (34)
60–70 years		428 (28)
>70 years		573 (38)
Male sex		860 (57)
ECOG	1 (0–4)	
0–1		789 (79)
≥2		210 (21)
WBC count (×10^9^/L)	8.6 (0.1–401)	
Hemoglobin (g/dL)	8.9 (2–15.9)	
Platelet count (×10^9^/L)	54 (3–907)	
BM blast %	54 (1–100)	
Cytogenetic risk MRC (n = 824)		
Low		68 (8)
Intermediate		485 (59)
High		271 (33)

Abbreviations: MRC, Medical Research Council; AML, acute myeloid leukemia; BM, bone marrow; WBC, white blood cell count.

**Table 2 cancers-17-00854-t002:** NGS and PCR results for each mutation, with NGS VAF cut-off ≥5%.

Gene Mutation		PCR+ N(%)	PCR− N(%)	TOTAL
*NPM1*	NGS+	262 (98.5)	10 (1.1)	272
	NGS−	4 (1.5)	880 (98.9)	884
	TOTAL	266 (100)	890 (100)	1156
*FLT3*-ITD	NGS+	149 (73.8)	4 (0.04)	153
	NGS−	53 (26.2)	1061 (99.6)	1114
	TOTAL	202 (100)	1065 (100)	1267
*FLT3*-TKD	NGS+	49 (84.5)	8 (0.07)	57
	NGS−	9 (15.5)	1064 (99.3)	1073
	TOTAL	58 (100)	1072 (100)	1130
*IDH1*	NGS+	77 (98.7)	11 (1.3)	88
	NGS−	1 (1.3)	836 (98.7)	837
	TOTAL	78 (100)	847 (100)	925
*IDH2*	NGS+	107 (99.1)	19 (2.3)	126
	NGS−	1 (0.9)	804 (97.7)	805
	TOTAL	108 (100)	823 (100)	931

Abbreviations: NGS, next generation sequencing; PCR, polymerase chain reaction; FLT3, FMS-like tyrosine kinase-3.

**Table 3 cancers-17-00854-t003:** NGS and PCR results for each mutation using NGS VAF cut-off ≥1%.

Gene Mutation		PCR+ N(%)	PCR− N(%)	TOTAL
*NPM1*	NGS+	264 (99.2)	14 (1.6)	278
	NGS−	2 (0.8)	876 (98.4)	878
	TOTAL	266 (100)	890 (100)	1156
*FLT3*-ITD	NGS+	176 (87.1)	11 (0.1)	187
	NGS−	26 (12.9)	1054 (99)	1080
	TOTAL	202 (100)	1065 (100)	1267
*FLT3*-TKD	NGS+	53 (91.4)	34 (3.2)	87
	NGS−	5 (8.6)	1038 (96.8)	1043
	TOTAL	58 (100)	1072 (100)	1130
*IDH1*	NGS+	77 (98.7)	18 (2.1)	95
	NGS−	1 (1.3)	829 (97.9)	830
	TOTAL	78 (100)	847 (100)	925
*IDH2*	NGS+	107 (99.1)	22 (2.7)	129
	NGS−	1 (0.9)	801 (97.3)	802
	TOTAL	108 (100)	823 (100)	931

Abbreviations: NGS, next generation sequencing; PCR, polymerase chain reaction; FLT3, FMS-like tyrosine kinase-3.

**Table 4 cancers-17-00854-t004:** FLT3-TKD alterations identified by NGS (cut-off ≥1%) and not detected in PCR analysis.

Protein Change Detected	Number of Patients	VAF (%) Median (Range)
Asp835Tyr	14	3.2 (1–44.29)
Asp835His	6	2.1 (1.23–46.6)
Asp835Glu	5	1.9 (1.13–4.68)
Ile836del	5	3 (1.91–12.53)
Asp835Val	2	21.43 (1.4–41.46)
Asp835Ile	1	3.11
Asp835del	1	4.3

Abbreviations: VAF, variant allele frequency.

## Data Availability

The original contributions presented in this study are included in the article/Appendix A. Further inquiries can be directed to the corresponding authors.

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
