# Peer review of "Conventional PCR Versus Next Generation Sequencing for Diagnosis of FLT3, IDH and NPM1 Mutations in Acute Myeloid Leukemia: Results of the PETHEMA PCR-LMA Study"

_cancers, 2025, doi:10.3390/cancers17050854_

Round 1
Reviewer 1 Report
Comments and Suggestions for Authors
The manuscript by Boluda et al. compares the performance of PCR and NGS in detecting specific mutations in AML patients. While the data presented here are not novel, they have the advantage of reflecting real-life clinical scenarios.
Throughout the manuscript, the performance of NGS is evaluated using PCR as the gold standard. This approach results in a seemingly low sensitivity for NGS in detecting FLT3-TKD and FLT3-ITD mutations. However, although it is well established that NGS often fails to detect true positive variants for FLT3-ITD, it typically performs better for FLT3-TKD—a point that is not clearly highlighted in the manuscript. The authors could improve clarity by explicitly defining what they consider as false/true positive/negative results, particularly for FLT3-TKD and IDH1/IDH2 mutations. For instance, the NGS analysis with a 1% VAF cut-off identifies more positive results than conventional methods: are these additional mutations true positives? If so, the authors should address this in the conclusion. Specifically, do they recommend NGS over conventional PCR for FLT3-TKD and IDH1/IDH2 mutations? Currently, only FLT3-TKD is mentioned in the conclusion.
Moreover, the authors employ two different VAF cut-offs for NGS detection (1% and 5%), which result in noticeably different performance outcomes. What is the rationale for including both thresholds in the manuscript? The authors could strengthen their conclusions by clarifying whether they consider mutations detected with the 1% VAF cut-off to be true positives. If so, do they recommend adopting the 1% cut-off as the standard?
The quality of the figures, particularly Figures 4 and 5, could be improved for better readability and interpretation.
Minor correction: In the paragraph "Single gene testing methods" (page 3), it is stated that the analysis of NPM1 PCR products by fragment analysis shows that the "wild-type allele generates a 198 bp PCR product and the mutated allele generates a fragment of 203 bp." However, NPM1 mutations result in a +4 nucleotide difference, so there appears to be an error in the numbers provided (203 - 198 = 5 bp). This should be corrected.
Author Response
- Throughout the manuscript, the performance of NGS is evaluated using PCR as the gold standard. This approach results in a seemingly low sensitivity for NGS in detecting FLT3-TKD and FLT3-ITD mutations. However, although it is well established that NGS often fails to detect true positive variants for FLT3-ITD, it typically performs better for FLT3-TKD—a point that is not clearly highlighted in the manuscript. The authors could improve clarity by explicitly defining what they consider as false/true positive/negative results, particularly for FLT3-TKD and IDH1/IDH2 mutations. For instance, the NGS analysis with a 1% VAF cut-off identifies more positive results than conventional methods: are these additional mutations true positives? If so, the authors should address this in the conclusion. Specifically, do they recommend NGS over conventional PCR for FLT3-TKD and IDH1/IDH2 mutations? Currently, only FLT3-TKD is mentioned in the conclusion.
#Response to reviewer´s comment:
We agree that NGS usually performs better for FLT3-TKD mutation, but in our study, when lowering the VAF cut-off to 1%, only 5 discrepancies remain. For a better clarification, we have included a table to highlight the positive FLT3-TKD by NGS with negative result by PCR.
For the purposes of this study, we consider the positive detected with a 1% VAF cut-off as true positives and we have included in the discussion: “On the other hand, for FLT3-TKD, NGS missed 9 positive samples using a cut-off of 5%, probably because the sensitivity of PCR is 3%. Lowering the cut-off to 1% allowed additional positive samples to be identified. For the purposes of this study we consider these as true positives, and from a clinical point of view, FLT3 inhibitor would be included as part of the treatment.” We agree with the reviewer that this cut-off should be recommended for these mutations and we have included: “Therefore, we consider that NGS should replace PCR for FLT3-TKD and also IDH testing if the results delivery timelines are shortened for the purposes of target therapy establishment” in discussion and “For these mutations with high clinical impact, a VAF cut-off of 1% should be adopted” in conclusion. We have also clarified that NGS is the preferred method for both FLT3-TKD and IDH mutations.
- Moreover, the authors employ two different VAF cut-offs for NGS detection (1% and 5%), which result in noticeably different performance outcomes. What is the rationale for including both thresholds in the manuscript? The authors could strengthen their conclusions by clarifying whether they consider mutations detected with the 1% VAF cut-off to be true positives. If so, do they recommend adopting the 1% cut-off as the standard?
#Response to reviewer´s comment:
We have employed two different VAF cut-offs for NGS detection in order to explore performance outcomes, and we recommend adopting the 1% cut-off as the standard for these mutations. We have included in the discussion: “In this study, we have employed two different VAF cut-offs for NGS detection in order to explore performance outcomes, which result in noticeably different results. For these mutations with high clinical impact, we consider these results as true positives and they are thus informed and considered. We recommend adopting the 1% cut-off as the standard for these mutations and their clinical implications.”
- The quality of the figures, particularly Figures 4 and 5, could be improved for better readability and interpretation.
#Response to reviewer´s comment:
As requested by the reviewer, we have modified these figures.
- In the paragraph "Single gene testing methods" (page 3), it is stated that the analysis of NPM1 PCR products by fragment analysis shows that the "wild-type allele generates a 198 bp PCR product and the mutated allele generates a fragment of 203 bp." However, NPM1 mutations result in a +4 nucleotide difference, so there appears to be an error in the numbers provided (203 - 198 = 5 bp). This should be corrected.
As requested by the other reviewer, we have simplified the Methods section for a better readability and corrected this.

Reviewer 2 Report
Comments and Suggestions for Authors
The study is clinically relevant, well-structured, and methodologically robust, but minor edits for clarity, readability, and interpretation of results would strengthen it. Some of the points are given below:
1. Some sections, especially the Methods and Results, contain dense technical descriptions and long sentences, making readability challenging. Consider simplifying or breaking up complex information for better reader engagement.
2. The figures and tables are highly data-dense. Can the authors add a graphical abstract or summary table highlighting key findings?
3. FLT3-ITD sensitivity in NGS was only 73.8%, indicating that some clinically relevant mutations might be missed. The authors could add a brief discussion on how missing these mutations could impact patient stratification for FLT3 inhibitors (e.g., midostaurin, gilteritinib).
4. The turnaround time (PCR = 7 days, NGS = 28 days) is one of the study’s most practical takeaways. It would be useful to discuss whether newer, more rapid NGS technologies (e.g., nanopore sequencing) could bridge this gap in the future.
Author Response
Comments and Suggestions for Authors
The study is clinically relevant, well-structured, and methodologically robust, but minor edits for clarity, readability, and interpretation of results would strengthen it. Some of the points are given below:
- Some sections, especially the Methods and Results, contain dense technical descriptions and long sentences, making readability challenging. Consider simplifying or breaking up complex information for better reader engagement.
#Response to reviewer´s comment:
As recommended by the reviewer, we have simplified the Methods section, and reorganized Results (Section 3.5 and 3.6) in additional subheadings for better readability.
- The figures and tables are highly data-dense. Can the authors add a graphical abstract or summary table highlighting key findings?
#Response to reviewer´s comment:
We thank the reviewer the suggestion but as there are already many figures, we prefer to include a graphical abstract if this will be published apart (relying to the journal policy).
- FLT3-ITD sensitivity in NGS was only 73.8%, indicating that some clinically relevant mutations might be missed. The authors could add a brief discussion on how missing these mutations could impact patient stratification for FLT3 inhibitors (e.g., midostaurin, gilteritinib).
#Response to reviewer´s comment:
We have reorganized the discussion to emphasize how missing these mutations could impact patient stratification for FLT3 inhibitors: “Regarding FLT3-ITD detection, we found low sensitivity of NGS due to limitation that this technique has for detecting duplications with long fragments [32–34]. The use of VAF ≥1% cut-off can improve detection efficiency but still persisted a 12% of false negative results with NGS. As a consequence the detection of FLT3-ITD mutation by NGS could be underestimated [35]. Therefore we believe that nowadays NGS cannot still replace PCR for FLT3-ITD screening, since missing these mutations could impact patient stratification for FLT3 inhibitors (e.g., midostaurin, gilteritinib).”
- The turnaround time (PCR = 7 days, NGS = 28 days) is one of the study’s most practical takeaways. It would be useful to discuss whether newer, more rapid NGS technologies (e.g., nanopore sequencing) could bridge this gap in the future.
#Response to reviewer´s comment:
In the last paragraph of discussion we address this issue and the need of advances: “We can imagine a not-too-distant future in which, for the sake of efficiency, NGS will replace conventional PCR, at least for the vast majority of relevant mutations. The use of NGS for routine clinical care will require new NGS technologies (e.g., nanopore sequencing), more robust bioinformatic tools to improve detection of FLT3-ITD mutations, the ability to detect a wide range of alterations (including rearrangements i.e MLLr) [36], and automated workflow to deliver a rapid turnaround time. These technological advances could pave the way to have a fully NGS-based molecular diagnosis in acceptable timelines (<5-7 days) allowing for urgent personalized medicine proper of AML.”
